# From road centrelines to carriageways—A reconstruction algorithm

**Stelios Vitalis**[ID]<sup></sup>*, **Anna Labetski**, **Hugo Ledoux**‡, **Jantien Stoter**‡

3D Geoinformation Group, Delft University of Technology, Delft, The Netherlands

 These authors contributed equally to this work.
‡ HL and JS also contributed equally to this work.
* s.vitalis@tudelft.nl

**Data Availability Statement:** All data used in this study are available as open data. The source code of our implementation is available in a public git repository (https://github.com/tudelft3d/carriageways-creator). The DOI to our resulting

## Abstract

Roads are important for many urban planning applications, such as traffic modelling and delivery vehicle routing. At present, most available datasets represent roads only as centrelines. This is particularly true for OpenStreetMap which provides, among many features, road networks at worldwide coverage. Furthermore, most approaches for creating more detailed networks, such as carriageways or lanes, focus on doing so from sources that are not easy to acquire, such as satellite imagery or LiDAR scans. In this paper we present a methodology to create carriageways based on OpenStreetMap's centrelines and open access areal representations (i.e. polygons) to determine which roads should be represented as two individual carriageways. We applied our methodology in five areas across four different countries with different built environments. We analysed the outcome in a delivery routing problem to evaluate the validity of our results. Our results suggest that this method can be effectively applied to create carriageways anywhere in the world, as long as there is sufficient coverage by OpenStreetMap and an areal representation dataset of roads.

## Introduction

Roads are a significant aspect of the built environment and are an important consideration for urban planners. They are an essential input in a variety of applications including urban traffic modelling [1], cycle accident analysis [2], vehicle routing [3], and municipal road maintenance, including: de-icing, weed control, road markings and road lighting [4].

Roads are usually modelled as networks at various representation levels (Fig 1): centrelines, carriageways, and lanes [5]. [6] examined various road applications and evaluated the representation level required for analysis. The results of the analysis indicated that there is no "one-size-fits-all" solution to road modelling. This is inline with other city objects, like buildings [7]. Therefore, an important aspect of designing as solution for a particular application consists of selecting or preparing data at the appropriate representation level.

There are several data generation methodologies in existence for the modelling of roads. For most use cases, the automatic reconstruction of road networks has mainly focused on

datasets is: https://doi.org/10.6084/m9.figshare.18550802.v2.

**Funding:** The research leading to this paper has received funding from the European Research Council under the European Union's Horizon2020 ERC Agreement no. 677312 UMnD: Urban modelling in higher dimensions which was granted to JS. The funders had no role in study design, data collection and analysis, decision to publish, or preparation of the manuscript.

**Competing interests:** The authors have declared that no competing interests exist.

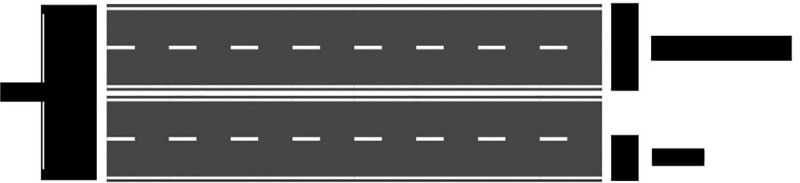

**Fig 1. The various portions of a road as can be represented from an areal view.**

generating the centrelines of roads, often from satellite imagery (see Section Procedural generation of road data). While for many applications this is sufficient, there are certain cases—such as road maintenance or traffic simulation—where this is not enough. For these applications, carriageway representation is required for a meaningful analysis.

In contrast to carriageways, there is an abundance of existing datasets of road centrelines, both within governmental spatial data infrastructures and as open data (e.g. OpenStreetMap). To satisfy the requirements of use cases as stated above, there is an increased push for developing multi-representation datasets for roads, such as for institutions in the Netherlands [8], which can be created through an automated process.

In this paper we present a novel reconstruction algorithm for deriving an approximation of carriageways from centrelines (Section Automatically reconstructing carriageways from a combination of linear and areal roads). The methodology creates carriageways for centrelines that fit two flows of traffic and also focuses on reconstructing the network at intersections, especially complex ones, that are not trivial to create. We implemented our methodology in the Python programming language that runs as a Jupyter notebook (Section Implementation). Our methodology utilises the popular open data source of OpenStreetMap which makes our approach geographically agnostic and allows us to reconstruct carriageways of roads anywhere in the world. We tested our prototype with datasets from The Hague, Netherlands; Helsinki, Finland; Poznań, Poland; Shawinigan, Canada; and Toronto, Canada (Section Analysis of the reconstructed carriageways network). Both our code and data are openly available. Finally, we validate the quality of our reconstruction by using the network in a routing algorithm (Section Network analysis).

## Related work

### Procedural generation of road data

There are only a few studies that address carriageway-level modelling and a further few that address lane-level modelling. The majority of literature in the field of generating road datasets focuses on generating the datasets from satellite and aerial images. The focus is often on generating centrelines from road polygons, extracting the shape of roads, and/or building a road network with edges and nodes [9–11]. There is also a large body of work on reconstructing road networks based on GPS traces/trajectories [12–14]. GPS can also be used in combination with dead reckoning (calculating the current position of a moving object by using a previously determined position), inertial navigation systems, cameras and/or LiDAR [15, 16]. One study combined the acquisition trajectory from an autonomous vehicle with a professionally surveyed road network to derive a lane-level network [17]. A similar study harnessed OpenStreetMap (OSM) with local sensor information from 3D LiDAR as well as a positioning system to generate a lane-level dataset [18].

## Roads in 3D city models

From a roads modelling perspective, there is an overlap with the 3D city models domain on investigating and evaluating the practicality of multiple representations of road networks in one dataset. Within the realm of 3D city models, the concept of modelling features at various levels of representation has been heavily investigated, and is referred to as the Level of Detail (LoD). Furthermore, 3D city models support the storage of roads as areal or linear features in multiple LoDs. This is defined in both CityJSON [19] and CityGML [20], the most popular 3D city models standards. Therefore, we can draw a direct mapping between road network representation type (i.e. centrelines, carriageways and lanes) and the LoDs of roads in 3D city models.

While both standards define the concept of LoDs, the storage of multiple LoDs of roads in one dataset is an ongoing development and is being refined to address the needs of practitioners [5]. The work of [21] recommends for LoDs to be driven more by road surface semantics than geometry, this includes footpaths, road markings, road damage, etc. [5] instead focused on a more strictly geometric definition for LoDs, and linking them to modelling transportation at the road, carriageway, and lane level.

Based on this refined concept of LoDs, [6] examined applications by evaluating the appropriate LoD representation required for different analyses, see Table 1. The work examined both areal and linear representations and found that different applications had varying needs, meaning that a one-size-fits-all road dataset is not realistic.

## OpenStreetMap

OpenStreetMap (OSM) is open-access data that is generated by volunteers who contribute and maintain data about roads, railway stations, and various points of interest, around the globe [22]. The data does not stop at country borders and therefore OSM supports a generic modelling approach. OSM is based on crowdsourcing geospatial information and thereby opens the domain to all users without restricting it to the traditional cartographers or geographers [23].

Features in OSM are referred to as `elements`. `Elements` are of three types: `nodes` (points), `ways` (linear features and area boundaries), and `relations` (representations of how elements relate to each other) [24]. `Elements` can have one or more associated `tags`, these describe the meaning of a particular `element`, i.e. the attributes. There are guidelines

**Table 1. Road data representation level required by potential applications.** Table based on the work of [6].

| | Linear | | | Areal | | |
|---|---|---|---|---|---|---|
| | Centrelines | Carriageways | Lanes | Centrelines | Carriageways | Lanes |
| Road repair | | | | | | x |
| De-icing roads | | x | x | | x | x |
| Disaster management | | | x | | | x |
| Surface heat monitoring | | | | | x | x |
| Air quality monitoring | | | x | x | x | x |
| Visibility analysis | | | | | x | x |
| Noise mapping | | | x | x | x | x |
| Traffic light configuration | | | x | | | x |
| Traffic simulations | | x | x | | x | x |
| Routing / navigation | x | x | x | | | |
| Autonomous driving | | | x | | | x |

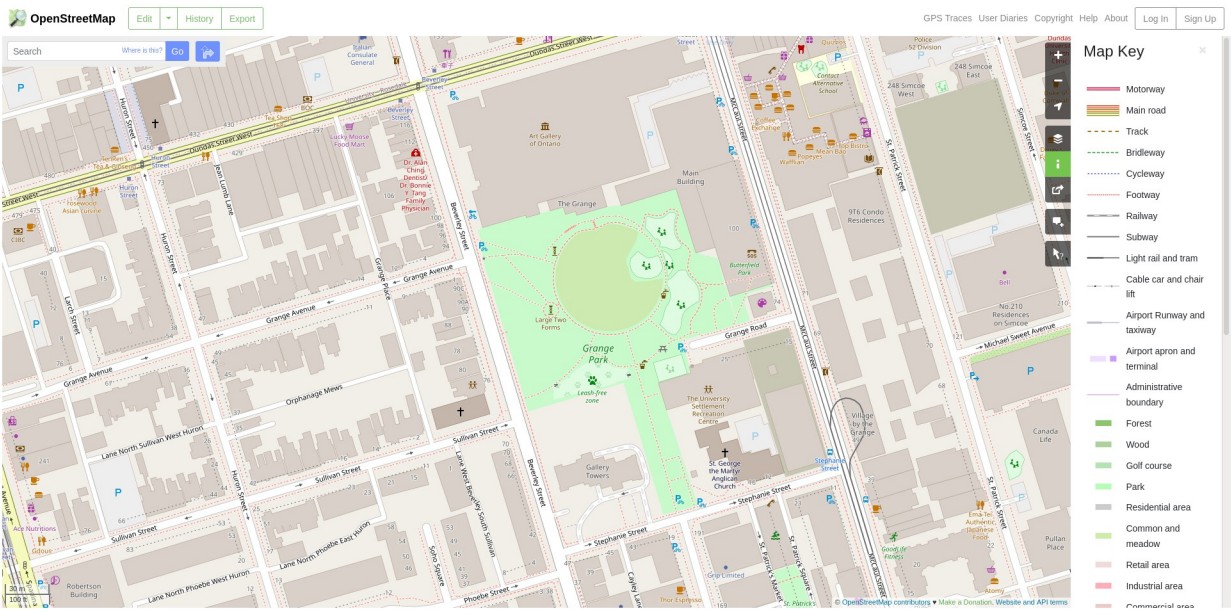

**Fig 2. Screenshot of the OpenStreetMap web viewer.** The data is visualised based on the semantics that are derived from the elements tags. For example, road width is implied based on the road's type, so the map renders these lines with the respective thickness. (Reprented from OpenStreetMap under a CC BY License, with permission from OpenStreetMap, original copyright OpenStreetMap contributors.).

about how attributes should be used to semantically mark the elements; for instance, a `way` is considered a road when the `tag` "highway" is set to a value from a predefined list of road types. The semantics can be used, then, to make a styled map for visualization purposes deriving, for instance, the colour and width of roads from their attributes (Fig 2). In case of areal objects (such as buildings and green spaces) the area is represented by a closed `way` that defines its boundaries and the `tags` of the `way` are used to describe the semantics and attributes of the area.

OSM is frequently compared to commercial and governmental datasets, and due to the active maintenance and regular contribution efforts of OSM volunteers, it is often faster to reflect real-world changes and tends to have more information. In a German road network study, it was found that the difference between the OSM date and a comparable proprietary dataset was only 9% [25]. Furthermore, the analysis uncovered that OSM data exceeded the information provided by the proprietary dataset by 27% [25]. They identified that there is OSM is actively utilised with an active community generating software [26–28], assessing errors [29–31], and applying it in use cases [32–34].

## Automatically reconstructing carriageways from a combination of linear and areal roads

Our objective is to reconstruct the carriageway representation of roads from centrelines. We require as input data: a) the linear representation of the centrelines of a transportation network (i.e. road lines from OSM), and b) the areal representation of a transportation network (i.e. polygons). The most important attributes necessary for identifying the presence of carriageways are: a) the OSM attribute for two-way traffic, and b) the road width, which we calculate as a part of our methodology using the areal representation. An overview of the process can be seen in Fig 3.

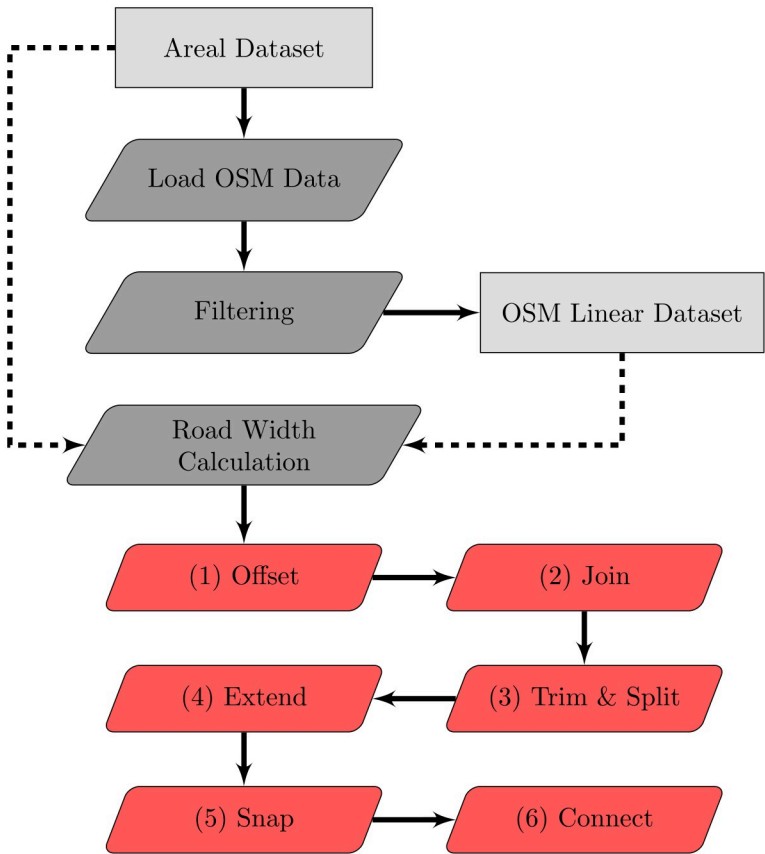

**Fig 3. The main structure of the reconstruction methodology.** Light grey rectangles represent data; dark grey trapeziums represent preprocessing steps; red trapeziums represent the iterations of our the main methodology.

## Important terminology

**Road** A real world feature with surfaces that facilitate transportation, primarily for automotive vehicles. A road spans from intersection to intersection.

**Road segment** A part of the road that might distinguish itself from the rest of the roads due to some property (e.g. width or transportation type).

**Linear Representation**[*] A transportation dataset with roads represented as a network.

**Areal Representation** A transportation dataset with the traversable portion of roads represented as a polygon with measurable areas and widths.

**Centreline (CL)** An edge of a network that represents the centreline of a road segment. In our case, this is the OSM data.

**Carriageway (CW)** An edge of a network that represent the individual streams of traffic flow in a road. Dual carriageways allow for bi-directional traffic in two different flows, while single carriageways either allow traffic flow in one direction, or they allow bi-directional traffic through the same stream.

**Intersection** Any point in a transportation network that intersects with more than two roads. We consider intersections to be nodes of the network of centrelines that have a degree of three or higher.

**Edge** In a network, an edge is an independent linear feature within the dataset. The edge represents a road segment as a centreline or a carriageway.

**Node** In a network, a node is a vertex incident to one or more edges.

**Degree of node** Represents the number of incident edges of nodes. For instance, a node that is incident to three edges is of degree three.

**Neighbours** For an edge *A*, neighbours are considered any edges that share an incident node with *A*. An edge should have at least one neighbour, otherwise it is not connected to the main network.

**Weakly connected component** In a network, a weakly connected component represents a subset of the network for which every node can reach at least one other node of this subset.

* Representation can be for centrelines or carriageways.

**Filtering.** As described in Section, OSM data is composed of `elements` of which roads are just a subset. Specifically, road segments are represented as linear `elements` among others, such as building footprints and land use areas (which are represented as closed polylines). To extract only the roads, we filter the original data so that we keep only those lines that have a value of the `highway` tag indicating that it is some form of a motorway. The following values of `highway` were kept:

- primary
- secondary
- motorway
- trunk
- tertiary
- unclassified
- residential
- motorway_link
- trunk_link
- primary_link
- secondary_link
- tertiary_link
- living_street
- service
- pedestrian
- bus_guideway
- escape
- raceway
- road

**Table 2. Connected components within the OSM data per area.**

|  | Number of Connected Components | Percent of edges within the largest Connected Component |
|---|---|---|
| The Hague | 96 | 99.2% |
| Helsinki | 376 | 97.5% |
| Poznań | 106 | 99.2% |
| Shawinigan | 6 | 98.1% |
| Toronto | 56 | 99.7% |

After we ensure the data contain only the types of roads that we want to investigate, we further refine the area of the network spatially. This is because the original data is downloaded based on the bounding box of the areal representation boundaries. In order to minimize the road features that are out of the area of interest, we compute the convex hull of the areal representation and select only the road features that intersect with it.

To ensure that the filtered roads compose a connected network, we do a basic network analysis to compute the individual weakly connected components. All edges within a connected component can reach each other and are therefore connected. Given that we extract data based on a convex hull, it is likely that we may inadvertently extract roads that are not connected to any/many other roads. We noticed that in all five areas we analysed, the largest weakly connected component contains over 97% of the nodes (see Table 2) therefore we filter the network further to exclude the edges and nodes that do not belong to it.

## Road width calculation

OSM stores as an attribute whether an edge permits bi-directional traffic. This however does not automatically mean that the road segment that the centreline represents has the space for two physical carriageways. Especially in Europe, there are many narrow roads in low-traffic areas where the road should still be modelled as a single carriageway despite allowing for bi-directional traffic. We believe that excluding these types of edges from the carriageway reconstruction (i.e. keeping them as simple centrelines) is an approach that is more aligned with the real-world representation and better mimics actual driving conditions. Therefore, road width calculations per centreline are necessary to determine whether a road segment is wide enough to be considered a dual carriageway. We use the logic that lanes tend to be 3.5 meters wide and therefore a dual carriageway road should be approximately at least 7 meters wide.

We calculate the road width based on the areal representation datasets according to the work of [35]. The approach was developed specifically for the BGT (Dutch: Basisregistratie Grootschalige Topografie, English: Key Register Large-Scale Topography) as areal representation of the Netherlands and NWB (Dutch: Nationaal Wegenbestand, English: National Road Database) as linear representation. We adapted it to run with the linear OSM data and any areal input file.

The algorithm segments every edge in the OSM input data into 11 parts of equal length. At each cut a 'measuring line' is created transversely to the longitudinal direction, which results in 10 measuring lines per edge (Fig 4). The width of the road at this location is determined by the length of the intersection of the measurement line with the road polygons that it intersects. Four values are calculated for each edge: mean, minimum, and maximum width, as well as the standard deviation. Then, the final width is computed by excluding outliers based on the standard deviation and only computing the mean of the values inside this range.

This algorithm takes into account that there are cases where the centrelines and the polygons might not overlap perfectly (Fig 5). To overcome this issue, the algorithm always chooses

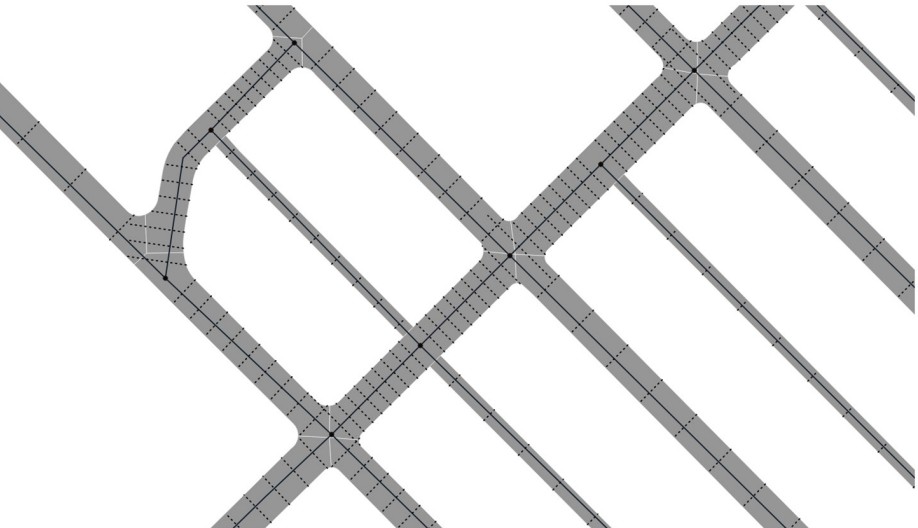

**Fig 4. Lines used to calculate the road width (excerpt of Toronto).** For every centreline (black solid line) there are nine perpendicular lines (black thin dotted lines) that are created, which are intersected with the polygons of the roads so that their length can be used to measure the width at the specific point.

the measuring line that is closer to the centreline, ignoring if it intersects the original centreline or not. We identified that for our datasets the percentage corresponds to the combined length of centrelines inside polygons related to the overall length of centrelines that overlap with the underlying polygons is around 70% (except for Helsinki where the overlap is only at 54% due to many small roads missing from the areal representation). Therefore, this is an important feature of the algorithm that allows us to have a robust output of road width calculation.

## Reconstruction

We generate a carriageways dataset by iterating over the edges of the network six times (Fig 6). During each iteration, we process and deal with one edge at a time and make specific decisions based on the edge and its neighbours. When we do an iteration, the algorithm is only aware of the current status of the edge being processed and the status of its neighbours prior to commencing the iteration. We ignore any transformation that may have already altered the adjacent edges within the current iteration. Not only does this speed up the processing time, but it also ensures that the reconstruction result is not influenced by the processing order of edges. Therefore, all edges are processed with the same logic based on a specific iteration step.

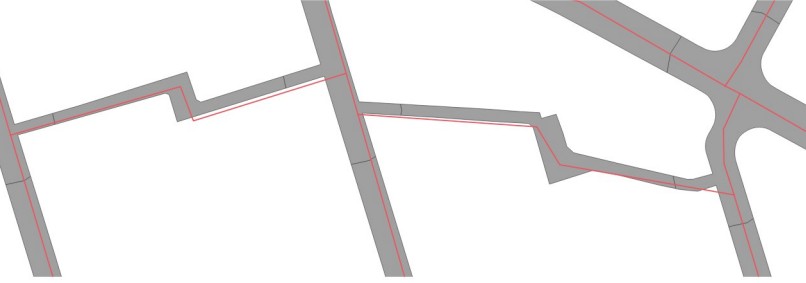

**Fig 5. Example of an imperfect overlap between the areal and linear datasets in Poznań.** Centrelines and polygons sometimes only partially overlap. This does not affect the road width calculation, as the closest perpendicular lines will be used regardless of the centreline's overlap with the polygon.

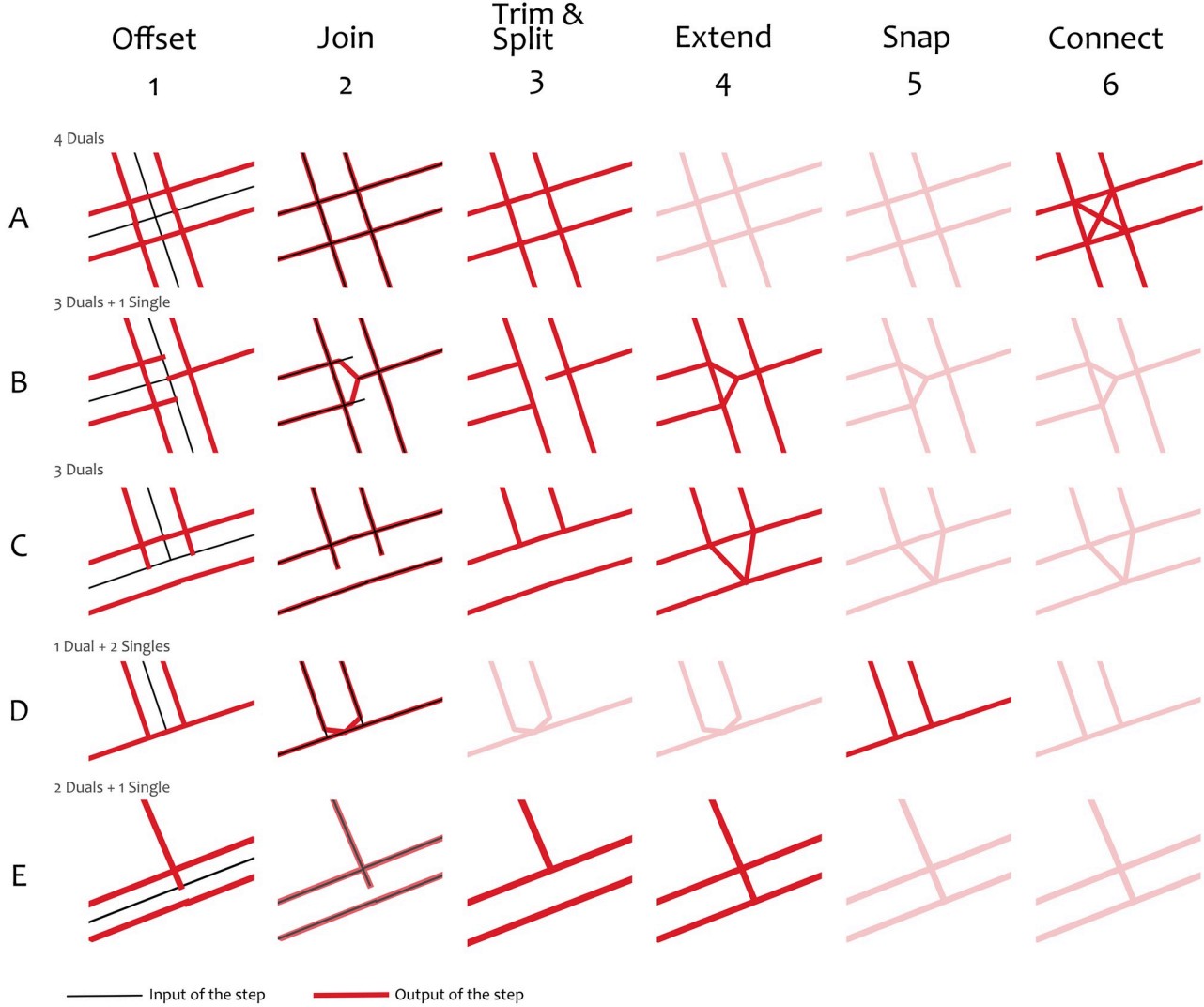

**Fig 6. Common intersection types in linear networks and their state during the six iterations of our methodology.** The rows represent 5 different cases (taken from OSM). *Note*: the inputs of steps 3 through 6 are omitted for the matter of simplicity.

**First iteration—Offset.** In this step the basic carriageways' geometry is created by offsetting the centrelines based on their directionality and the road width calculated (Section). Therefore, an edge can only be classified as a dual carriageway if it is bi-directional (based on its attributes) *and* wide enough. We assume that lanes are 3.5 meters wide and therefore a dual carriageway should be around 7 meters or more. We relax this rule to 6 meters given that there are various approaches to modelling the areal representation of roads. Specifically the presence of traffic islands is often recorded as holes in the datasets which reduces the width calculation.

We use the *adjusted mean* (i.e. mean excluding outliers) calculation of the road width to determine if a road segment is at least 6 meters wide and use an upper threshold of 15 meters. The latter was necessary because high values were found to occur in two situations. First, in many polygon datasets, intersections are modelled as one large polygon, and given that some datasets attributed large portions of a road segment to the intersection area, this created very wide polygons (see Section for details). Second, some datasets model parking areas or squares

as one large polygon which were not possible to filter due to inconsistent attribute names. Therefore, these were also accounted for with the upper threshold.

For edges that fall below the threshold of 6 meters or above the thresholds of 15 meters, we check the neighbouring edges to determine how to classify them. If both neighbours are classified as a dual carriageway then the edge is considered a dual carriageway and given the minimum average width between the two neighbouring road segments. This is only possible for edges that are not at intersections and for which we can assume a continuation of a road. We also deal with edges that are in between two intersections and have a road width beyond the upper threshold. If these edges with width less than 6 meters are less than 20 meters in length and are neighbouring at least one dual carriageway then they are also classified as a dual carriageway.

If an edge meets all the criteria described above then it is considered to be a dual carriageway. We then reconstruct the carriageways with two edges that are offset from the centreline based on the road width (Fig 6 *(Column 1)*). The carriageway edges are offset 25% of the road width in both directions from the road centreline. This is to model the carriageway edge as the centreline of the carriageway.

**Second iteration—Join.**   In this step, we join the endpoints of the new offset edges with the endpoints of their continuations. This is because edges that have been offset in the previous step will lose connectivity with their continuous neighbours for one of two reasons: either the neighbour is also a dual carriageway and, therefore, its offset endpoints will probably not match with the endpoints of the new offset because of their different in width; or the neighbour is a single carriageway, therefore it only has one endpoint which needs to be connected to the two endpoints of the carriageways. As a result, during this iteration we restore network connectivity between these edges. First, if the continuation of a dual carriageway edge is also a dual, then the midpoint between their two end points is calculated and the end points are adjusted to ensure that the two roads are now connected. If the continuation is a single carriageway then "forks" are created between the dual carriageway and the single carriageway. This is done by trimming the end points of the dual carriageway by 1 meter and joining the new end points to the end point of the single carriageway.

We have to clarify that a continuation is considered to be the neighbouring edge that is classified as the next edge of the same road. Therefore, if two edges are only linked with each other (via a 2-degree node) we count one as the continuation of the other. If an edge reaches an intersection, then we compute the continuation as the one that seems more likely to belong to the same road; this means that in most cases the edge with an angle closest to 180˚ is chosen, with only some exceptions where the directionality of road can be used to pick a better candidate. For instance, in an intersection of three roads were not all three edges have the same directionality we choose the link the two one-ways or two dual carriageways to be the continuation of each other.

**Third iteration—Trim and split.**   This step is used to trim or split parts of the edges that end up in intersection, in order to ensure that new nodes are added when the geometries of the new edges intersect. An intersection is classified as a node at which 3 or more centrelines meet. When a centreline becomes a dual carriageway, then its new edges are expected to intersect with other lines that it meets in the intersection (especially in perpendicular roads). Furthermore, this step is particularly important for intersections involving a mixture of dual and single carriageways. Dual carriageways are cut at the point at which they intersect with a road edge, the portion that is in the intersection is trimmed and removed because a different geometry will be introduced at the next step (Fig 6 *(B/C3)*). Edges representing single carriageways are either split or trimmed: when their continuation is a dual carriageway they are split at the point at which they intersect an edge because this will ensure that it is will still have network

connectivity with that edge (Fig 6 *(B3)*); in any other case, they are trimmed so that a new segment is going to be created in their place (Fig 6 *(E3)*).

**Fourth iteration—Extend.** In this step, new edges are being created to connect the parts that were previously trimmed in order to recover the connectivity of the network. In cases where three dual carriageways meet, a "fork" is formed between the three (Fig 6 *(C4)*). A "fork" is also formed when a dual carriageway becomes a single carriageway in an intersection (Fig 6 *(B4)*). A single carriageway that meets two dual carriageways also extends to the opposite carriageway (Fig 6 *(E4)*).

**Fifth iteration—Snap.** This step is to simplify the geometry of an intersection where a dual carriageway ends to a perpendicular single carriageway. In this case, we snap the "fork" pieces to the perpendicular road (Fig 6 *(D5)*). To do this, we identify the second-to-last point of the carriageway (i.e. the point at the base of the small "fork") and we compute its projection to the closest neighbouring edges in the intersection. Then, we move this node to its projection and we replace the last segment (which was previously a straight line) so that its geometry matches the same part of the neighbouring edge (e.g. if the line was ending to a curved intersecting line, then the last part of the new snapped edge will also follow the curve).

**Sixth iteration—Connect.** In this step, we add diagonal edges in four-way intersections of dual carriageways to ensure that traversing the new road carriageway network can better reflect realistic transport behaviour (Fig 6 *(A6)*). We, also, add new edges to connect the ends of the two carriageways of a road segment that were previously a dead end, to ensure that flow of traffic from one carriageway to another is possible.

## Addressing specific cases

The aforementioned methodology deals with all cases of intersections that occurred through the five datasets that we processed. Nevertheless, with each new dataset we learned of regional built environment differences that affected previous assumptions we made about our methodology. Therefore, for each of those we had to adopt our methodology accordingly, either adjusting our algorithm away from our previous false assumptions or by treating them as special cases. These cases (Fig 7) include:

1. Three dual carriageways meeting with non 90˚ angles (Fig 7 *(1)*), where identifying continuations have been proven relatively problematic. Initially, we assumed it would suffice to choose the line that is closest to the 180˚ for each of the edges involved. Nevertheless, this proved to cause issues in degenerate cases where the continuation of every edge was assigned to a single other edge and there was no single pair of edges that were marked as opposite continuations. In order to deal with this scenario, we compute continuations at two steps: first we pick candidates based on the initial logic (i.e. closer to straight next segments); then we ensure that we only allow an edge to pick another as a continuation as soon as the other also picks this as such. Finally, we wrote a specific logic to deal with 3-degree intersections of dual carriageways.

2. Two dual carriageways meeting at a 3-degree intersection with a 90˚ angle (Fig 7 *(2a-b)*). This case posed many challenges as to which decisions would make a useful outcome, from a geometric standpoint. This is because of two issues: first, the two dual carriageways had to be joined in such a way that their width is not altered (which was the case when we moved their endpoints to the middle of their distance); and second, the single carriageway should remain as much intact as possible, so the dual carriageways had to use it as a reference on how to be shaped. In order to achieve this, we decided that the outer side of the dual turn would have to construct an additional segment between one of the two dual roads and the

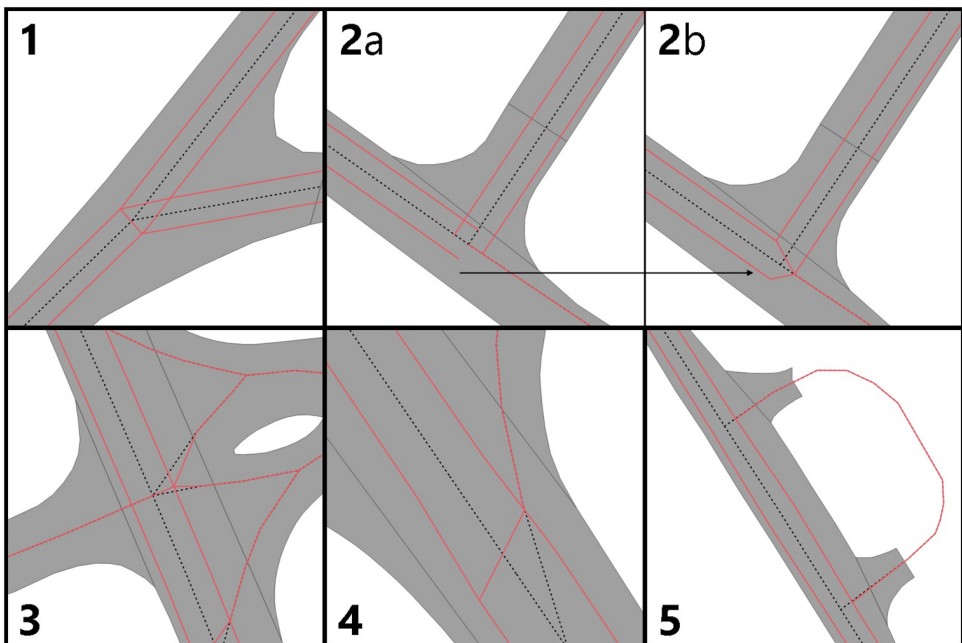

**Fig 7. Five special cases that affected our methodology: (1) three dual carriageways with unclear continuations; (2) a three-way intersection where two dual carriageways meet a single but the duals are perpendicular; (3) a dual carriageway that is crossing an existing merging dual carriageway from OSM; (4) exit lane from a dual carriageway; and (5) small segment that crosses the same road in both ends.**

intersection of the other dual road with the single one. This causes the single carriageway to move from its original endpoint, which is something we tried to avoid in most cases, but creates a more realistic output.

3. A created dual carriageway that crosses an dual carriageway already modelled in OSM (Fig 7 (3)), which is a special case of the general problem of dual carriageways that are already modelled as such in OSM. We did not find a trivial solution to identifying these OSM dual carriageways in order to treat them similarly to the dual carriageways constructed by our algorithm, which could have created a more realistic outcome in some cases. Therefore, we compromised with treating them as single carriageway roads. As shown in Fig 8 this might not seem as realistic as it could have, but it retains a network connectivity.

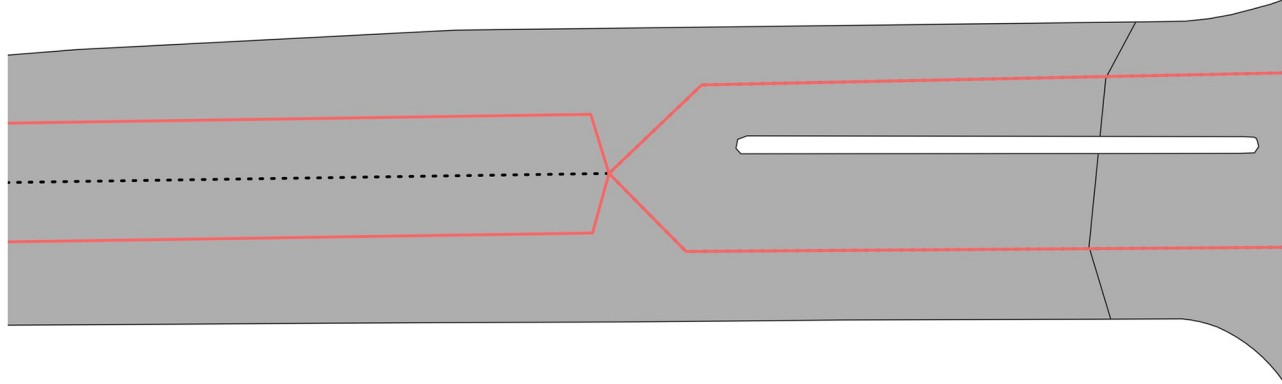

**Fig 8. Mismatch between existing dual carriageways in OSM and our reconstructed dual carriageways.**

4. Exit lanes from one carriageway of a dual road (Fig 7 (4)) are relatively unique both in matters of network topology (i.e. how they connect with the network) and geometry (their angle and the segments they intersect). In matters of topology, exit lanes are arbitrary regarding their connectivity with the other side of the road; should they cross the dual carriageway or not? In theory, OSM can contain such complex turning restriction information in its nodes, but due to the scope of the research we did not use them (neither did we find such information in some nodes that we investigated). While we could have tried to apply certain heuristics in order to classify exit lanes, we decided it would have been a complex and, possibly, unreliable process with little gain. Therefore, we chose to treat them as single carriageway edges that can always cross both sides of the road. Geometrically, exit lanes sometimes crossed the new carriageways at a later segment than the original one. This is due to their relatively small angle with the original road that they divert from, which means that if the original road segment is short, then the exit lane might cross the carriageway of the next segment instead. In our methodology, though, we were only testing for intersection of immediate neighbours, therefore those intersections where not detected leaving the exit lanes intact and breaking the network (as no node was created between the exit lane and the new carriageway). To overcome this, we added an extra rule where for short segments (we put a threshold of 7.5m) we also include their neighbours' neighbours geometry in the trimming and extending iteration.

5. Side roads that intersect the same dual carriageway road segment twice (Fig 7 (5)), were proven a degenerate case. Similarly to exit lanes, they were proven challenging both for topology and geometry. We chose to assume that these cases should be just trimmed and not allowed to cross both carriageways, because in most cases these where small segments which are expected to be just side roads and not big segments that allow for turns.

## Identification of errors

In order to validate our results and ensure a connected network we focused on identifying major errors in the final dataset. To achieve this, we used two metrics:

1. The number of weakly connected components (from now on, only referred as "connected components"), which is computed in order to ensure that either the whole network is fully connected, or that at least the overwhelming majority of edges belong to one component.

2. The number of false dead ends introduced in the carriageways network. In this context, false dead ends are nodes of the centrelines network that were of degree higher than two, but after the reconstruction they became dead ends (i.e. 1-degree nodes).

Table 3 lists the values of these metrics for the carriageways that we produced. While there were new connected components introduced into our network, the vast majority of edges remain in one main connected component. We also consider the percentage of false dead ends to be relatively low across all datasets (less than 10% of dead ends or 1% of all nodes).

**Table 3. Statistics related to errors.**

| Network Measures | The Hague | Helsinki | Poznań | Shawinigan | Toronto |
|---|---|---|---|---|---|
| *Number of connected components (cc)* | 29 | 17 | 13 | 17 | 213 |
| *Percentage of edges in the biggest cc* | 99.96% | 99.89% | 99.96% | 99.91% | 99.93% |
| *Number of false dead ends* | 187 | 316 | 198 | 76 | 912 |
| *False dead ends as a percentage of all dead ends* | 3.14% | 2.04% | 1.97% | 8.35% | 5.67% |
| *False dead ends as a percentage of all nodes* | 0.28% | 0.25% | 0.24% | 0.66% | 0.41% |

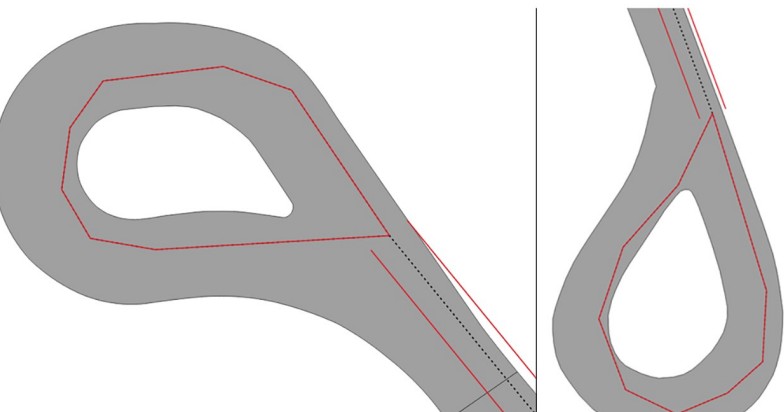

**Fig 9. Examples of loops at the end of roads in Toronto which result in false dead ends.**

Our investigation determined that the majority of small individual connected components and false dead ends are related to two cases: a) loops which are normally at the end of a road (Fig 9); and b) complex intersections that can be considered degenerate cases. We chose not to address these issues because loops do not significantly affect the overall connectivity of the network, while for complex intersections we only address cases to the extent to which our script can work with worldwide datasets and not degenerate cases that can occur due to the peculiarities of certain areas. In addition to these, the aforementioned metrics can highlight possible issues with the manipulation of geometries that can be caused due to limited precision of coordinates. This issue can affect operations such as the calculation of a point on a line, which can occur depending on the environment in which the notebook is run or due to limitations of the libraries used.

## Implementation

We implemented our methodology in Python in an openly available Jupyter notebook (Fig 10). Users only need to provide the areal representation for their area of interest. The road width calculation is done with a PostGIS script that is called directly from the Python script. The main libraries utilised are *geopandas* [36], *osmnx* [28], *networkx* [37] and *shapely* [38].

Once the areal representation of roads is loaded as a *geopandas* data frame, the convex hull is calculated in order to utilise *osmnx* to extract the OpenStreetMap (OSM) data for the same area. A UUID is computed for every edge in the OSM network.

The final step of our implementation generates metadata that documents all of the processing steps and input data. This ensures output data can be discoverable and analysed in fitness-for-purpose analysis based on a specific use case.

## Computational complexity

The computational complexity of the proposed methodology relies on the complexity of the width calculation and the individual iterations that compose it. This is as follows:

**Width calculation**: The width calculation iterates through all centrelines and uses a R-Tree index to identify neighbouring polygons. For simplicity, we assume that both centrelines and polygons are of magnitude $n$. Given the complexity of an R-Tree index lookup, which is $\mathcal{O}(\log n)$, the overall complexity of the loop is of $\mathcal{O}(n \log n)$ complexity.

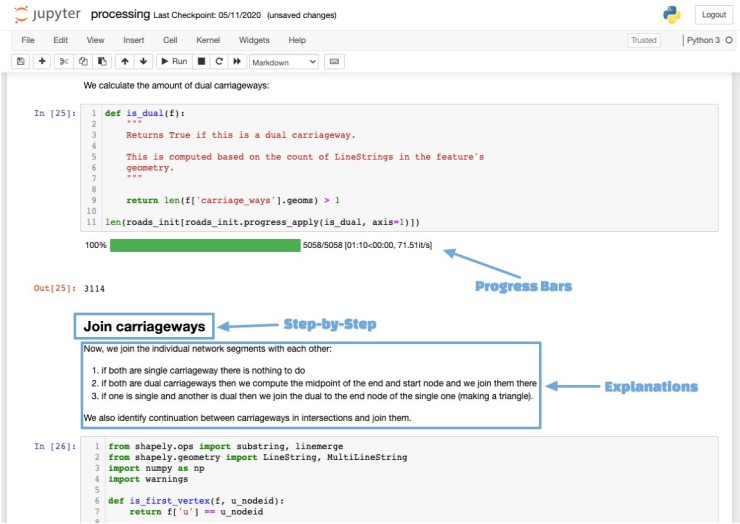

**Fig 10. View of the Jupyter output while running the script.**

**First iteration**: The first iteration is a loop over all the centrelines with no additional complexity per iteration, therefore the complexity of this step is $\mathcal{O}(n)$.

**Iterations 2-6**: All other iterations are a loop of $n$ centrelines in which every iteration relies on a lookup of neighbours in the network, which is of $\mathcal{O}(\log n)$ complexity. Therefore, the overall complexity of this step is $\mathcal{O}(n \log n)$.

Therefore the overall complexity of the methodology is:

$$\mathcal{O}(n \log n) \tag{1}$$

## Analysis of the reconstructed carriageways network

We tested our methodology with five different datasets across different regions. We processed The Hague, Netherlands; Helsinki, Finland; Poznań, Poland; Shawinigan, Canada; and Toronto, Canada. These were selected due to their differences in region and differences in size between the towns and cities. As the smallest area we have rural Shawinigan, with just over 5000 centrelines, and on the opposite scale we have urban Toronto with over 140,000 centrelines. With this diversity in areas we expect to have captured various different built environments and therefore vastly different road models. Although, it should also be said that choice was rather limited due to the lack of widely available open areal datasets.

Fig 11 presents the centrelines for the five datasets and their respective directionality graphs [39]. The five areas present different characteristics: The Hague has a relatively complex and irregular shape while maintaining a grid structure of roads; Helsinki has a more sparse network of roads and an equal distribution of roads orientation; Toronto is the densest area of all and as close to a grid as possible; Poznań is more similar to Helsinki regarding its density and orientation distribution, although it is slightly more irregularly shaped; and Shawinigan holds a more urban shape while portraying smaller grid structures which can be seen on its directionality graph.

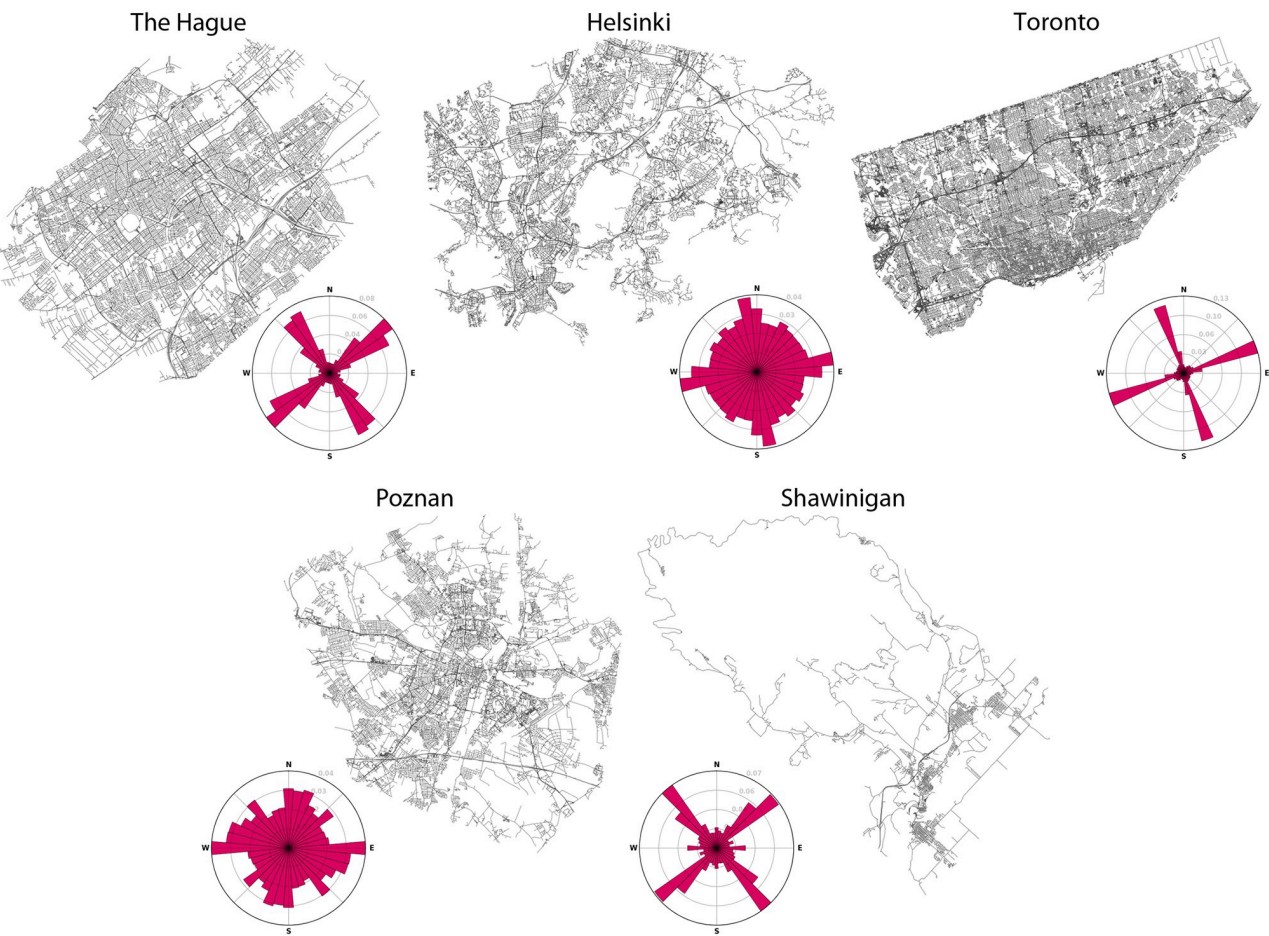

**Fig 11. Overview of the datasets analysed and their road orientation.** Overview of the datasets analysed and their roads' orientation graphs. Plots produced using [40]. (Reprented from OpenStreetMap under a CC BY License, with permission from OpenStreetMap, original copyright OpenStreetMap contributors.).

## Network metrics

Table 4 depicts various network measures to describe the input datasets, the carriageways we generated, and the differences between centrelines and carriageways. We used as our starting point the network measures as developed by [41].

The metrics allow us to see the impact of regional differences. It is evident that there are differences between Europe and Canada which can be observed from the proportion of dual carriageways. Toronto (36%) and Shawinigan (61%) have the highest proportions, which is unsurprising given the impact of cars on the development of Canadian cities. Furthermore, Shawinigan has the highest proportion of all; this is reflective of the space that such rural low-density communities have access to. Another value that highlights these regional differences is the road width calculation where the road width medians are much larger in Canada ($> 8.5m$) than in Europe ($< 7m$).

The metrics assist in comparing between the centrelines from OSM and the carriageways that we generated. A large difference between the two datasets is the change in length. While across all the datasets the total length increased, the lengths per edge actually decreased. This can be explained by the trimming and splitting that our reconstruction methodology relies on,

**Table 4. Network measures of the road datasets and comparisons between them.** Lengths and areas are measured in $m$ and $m^2$, respectively.

| | The Hague | Helsinki | Poznań | Shawinigan | Toronto |
|---|---|---|---|---|---|
| Centerlines (CLs) | | | | | |
| *Number of Edges* | 55450 | 93360 | 67194 | 5045 | 144942 |
| *Length—Mean* | 47.49 | 30.18 | 51.18 | 182.64 | 64.92 |
| *Length—Standard Deviation* | 64.53 | 45.61 | 76.77 | 361.74 | 84.44 |
| *Length—Median* | 29.69 | 16.25 | 26.45 | 93.66 | 39.14 |
| *Length—Total* | 2633566.97 | 2817813.7 | 3438957.8 | 921415.07 | 9410358.11 |
| *Edge Density* | 0.01596 | 0.01037 | 0.00806 | 0.00134 | 0.01287 |
| *Area to CLs Length Ratio* | 5.38 | 4.19 | 5.5 | 7.63 | 7.34 |
| *Number of Intersections* | 46001 | 85620 | 57920 | 3694 | 114095 |
| *Intersection Density* | 0.00028 | 0.00032 | 0.00014 | 0.00001 | 0.00016 |
| *Node Degree—Mean* | 3.21 | 3.14 | 3.13 | 3.26 | 3.17 |
| *Node Degree—Standard Deviation* | 0.43 | 0.37 | 0.36 | 0.45 | 0.39 |
| *Node Degree—Median* | 3 | 3 | 3 | 3 | 3 |
| *Node Degree—Mode* | 3 | 3 | 3 | 3 | 3 |
| *Number of Dead-ends* | 5405 | 14765 | 9383 | 564 | 12868 |
| Areal Data | | | | | |
| *Area of Convex Hull* | 165056810.8 | 271700741.4 | 426737497.5 | 688531584.5 | 731087991.7 |
| *Number of Polygons* | 55362 | 40572 | 14091 | 4443 | 66911 |
| *Polygon Area—Mean* | 256.08 | 290.96 | 1341.75 | 1582.44 | 1032.14 |
| *Polygon Area—Standard Deviation* | 560.89 | 1023.94 | 6002 | 4022.6 | 3364.6 |
| *Polygon Area—Median* | 121.6 | 10.45 | 415.43 | 968.75 | 567.7 |
| *Polygon Area—Total* | 14177035.12 | 11805016.15 | 18906653.51 | 7030788.4 | 69061449.35 |
| *Number of Polygons to Number of CLs Ratio* | 0.998 | 0.43 | 0.21 | 0.88 | 0.46 |
| *CLs to Polygon Overlap* | 72% | 54% | 68% | 63% | 74% |
| *Area to CLs Length Ratio* | 5.38 | 4.19 | 5.5 | 7.63 | 7.34 |
| Road Width | | | | | |
| *Road Width—Mean* | 10.48 | 8.97 | 8.69 | 9.07 | 10.35 |
| *Road Width—Standard Deviation* | 11.54 | 9.16 | 8.23 | 5.72 | 9.86 |
| *Road Width—Median* | 6.86 | 6.95 | 6.75 | 9.22 | 8.59 |
| *Road Width (Excluding Outliers)—Mean* | 7.34 | 8.17 | 7.68 | 9.55 | 8.86 |
| *Road Width (Excluding Outliers)—Standard Deviation* | 3.11 | 2.81 | 2.94 | 2.56 | 2.62 |
| *Road Width (Excluding Outliers)—Median* | 6.6 | 7.59 | 7.1 | 9.5 | 8.65 |
| *Number of CLs Classified as Two-way in OSM* | 34470 | 74360 | 52085 | 4454 | 120017 |
| *Proportion of Two-way CLs* | 62% | 80% | 78% | 88% | 83% |
| *Number of High Road Width Outliers* | 12456 | 21288 | 13142 | 495 | 37919 |
| *Proportion of High Road Width Outliers* | 22% | 23% | 20% | 10% | 26% |
| *Number of Low Road Width Outliers* | 10513 | 30831 | 16911 | 799 | 43161 |
| *Proportion of Low Road Width Outliers* | 19% | 33% | 25% | 16% | 30% |
| *Number of CLs Converted to Dual CWs* | 11775 | 25591 | 14422 | 3099 | 51881 |
| *Proportion of CLs Converted to Dual CWs* | 21% | 27% | 21% | 61% | 36% |
| *Proportion of Two-way CLs Converted to Dual CWs* | 34% | 34% | 28% | 70% | 43% |
| Carriageways (CWs) | | | | | |
| *Number of Edges in CWs Dataset* | 86079 | 146907 | 103027 | 17736 | 309394 |
| *CWs to CLs Ratio* | 1.55 | 1.57 | 1.53 | 3.52 | 2.13 |
| *CWs Length—Mean* | 36.96 | 24 | 40.42 | 84.27 | 44.79 |
| *CWs Length—Standard Deviation* | 57.6 | 40.5 | 69.09 | 268.74 | 75.11 |
| *CWs Length—Median* | 18.43 | 10.98 | 16.09 | 6.09 | 12.93 |

*(Continued)*

**Table 4.** (Continued)

| | The Hague | Helsinki | Poznań | Shawinigan | Toronto |
|---|---|---|---|---|---|
| CWs Length—Total | 3181560.22 | 3526219.63 | 4164786.21 | 1494554.89 | 13857188.88 |
| CWs Density | 0.01928 | 0.01298 | 0.00976 | 0.00217 | 0.01895 |
| CWs to CLs Total Length Difference | 547993.25 | 708405.94 | 725828.41 | 573139.82 | 4446830.77 |
| CWs to CLs Total Length Difference (%) | 21% | 25% | 21% | 62% | 47% |
| CWs to CLs Mean Edge Length Difference | -10.53 | -6.18 | -10.76 | -98.37 | -20.14 |
| CWs to CLs Mean Edge Length Difference (%) | -22% | -20% | -21% | -54% | -31% |
| Number of Intersections | 67315 | 124056 | 82484 | 11586 | 221199 |
| Intersection Density | 0.00041 | 0.00046 | 0.00019 | 0.00002 | 0.0003 |
| Node Degree—Mean | 3.37 | 3.42 | 3.37 | 4.01 | 3.68 |
| Node Degree—Standard Deviation | 0.53 | 0.55 | 0.54 | 0.66 | 0.66 |
| Node Degree—Median | 3 | 3 | 3 | 4 | 4 |
| Node Degree—Mode | 3 | 3 | 3 | 4 | 4 |
| Number of Dead-ends | 5958 | 15515 | 10072 | 910 | 16092 |
| CWs to CLs Intersections Change (%) | 146% | 145% | 142% | 314% | 194% |

as well as the introduction of small segments in the intersections. With respect to the nodes' complexity we noticed an increase in average nodes degrees for all datasets, but in different proportion between European and Canadian cities; all five datasets have a similar mean and median degree of around 3 in the original network, but the first ones only show a slight increase in mean, while the second showed a significant increase in node complexity moving their median up to 4. This is, also, underlined by the increase in number of intersections, where Shawinigan and Toronto showed a higher percentage of new intersections (i.e. nodes of 3 degrees or higher). We can link this to the differences in directionality between the areas (Fig 11) as cities with more irregular orientations are more prone to simple three-way intersections which are not often not altered during the creation of a dataset; in contrast, four-way intersections create many more nodes (e.g. four 4-degree nodes for an intersection where all incident roads create carriageways) which further skews the average.

Another interesting observation we made is that although the five cities do not differ greatly regarding the percentage of centrelines that are classified as two ways in OSM (mostly around 80%, except for The Hague), the same division between European and Canadian countries is evident with respect to the proportion of two-way centrelines that were converted to dual carriageways. Even more so, it is evident that Shawinigan demonstrates an outstanding increase in carriageways, compared to centrelines, possibly due to its rural setting. Although there is some correlation to the mean and median road width of the datasets and the final proportion of carriageways to centrelines, this cannot be linked directly as, for example, The Hague has a noticeably lower average road width than Helsinki and Poznań but they have very similar proportions of carriageways to centrelines. We believe that the urban nature of these three datasets and our heuristics that sometimes override the road width to avoid unnecessary dual carriageways must have been a significant factor there.

## Network analysis

In order to examine the difference between datasets and regions, we tested our results in a network analysis scenario, with grocery deliveries as a case study. Delivering groceries requires accurate routing models given that grocery goods are often perishable and must be delivered to customers at a designated time. Furthermore, a network analysis problem allows us to

**Table 5. Summary of the network analysis input parameters and results for centrelines and carriageways.**

| | The Hague | Helsinki | Poznań | Shawinigan | Toronto |
|---|---|---|---|---|---|
| Population | 514,861 | 631,695 | 540,365 | 50,060 | 2,930,000 |
| Number of Stores | 5 | 6 | 5 | 1 | 29 |
| Number of Buildings | 260,785 | 62,639 | 121,015 | 18,823 | 414,374 |
| Average Drive Distance—CL (km) | 3.05 | 4.00 | 5.39 | 7.46 | 2.63 |
| Average Drive Distance—CW (km) | 3.12 | 4.15 | 5.46 | 7.55 | 2.73 |
| Median Drive Distance—CL (km) | 2.88 | 3.90 | 5.12 | 7.21 | 2.52 |
| Median Drive Distance—CW (km) | 2.94 | 3.96 | 5.17 | 7.29 | 2.61 |
| Standard Deviation Drive Distance—CL (km) | 1.40 | 2.16 | 2.61 | 4.67 | 1.30 |
| Standard Deviation Drive Distance—CW (km) | 1.43 | 2.27 | 2.61 | 4.71 | 1.32 |
| Total Distance—CL (km) | 790,155.16 | 243,553.41 | 650,473.92 | 139,999.65 | 1,082,166.88 |
| Total Distance—CW (km) | 807,992.48 | 254,360.66 | 658,817.20 | 141,747.06 | 1,123,323.21 |
| Change in Total Distance (km) | 17,837.32 | 10,807.25 | 8,343.28 | 1,747.416 | 41,156.33 |
| Change in Total Distance (%) | +2% | +4% | +1% | +1% | +4% |
| Average Change in Distance (m) | 68.91 | 176.5 | 69.09 | 93.06 | 99.97 |
| Median Change in Distance (m) | 20.47 | 22.67 | 12.50 | -3.96 | 25.24 |
| Largest Decrease in Distance (km) | 3.23 | 2.96 | 1.24 | 2.36 | 3.67 |
| Largest Increase in Distance (km) | 4.53 | 5.98 | 4.06 | 2.95 | 3.34 |
| Shortest Route—CL (m) | 4.05 | 15.02 | 1.68 | 32.67 | 1.87 |
| Shortest Route—CW (m) | 4.82 | 15.01 | 1.68 | 18.21 | 5.42 |
| Longest Route—CL (km) | 8.93 | 11.18 | 15.04 | 28.60 | 9.86 |
| Longest Route—CW (km) | 9.07 | 11.81 | 15.05 | 46.09 | 9.87 |
| Number of Building-Store Allocation Changes | 2076 (0.80%) | 1666 (2.66%) | 1903 (1.57%) | - | 8595 (2.07%) |

validate the created carriageways network and to check for connectivity. We assume that there is roughly one large chain grocery store for 100,000 people [42] and based on the population we generated random points to act as grocery store locations. We accessed building outlines from OSM and generated a centroid per building to act as delivery locations. We ran a network analysis to see the fastest estimated drive time required per store to building. Only the biggest weakly connected component of the outcome of the reconstruction was used in the network analysis, to ensure that there is proper connectivity between all nodes. The analysis calculates the drive time from every store to every building but we filter so that we only analyse the shortest route for every building. We compared the results between the road centrelines and the carriageways (results in Table 5). Analysis was run with the QNEAT3 plugin in QGIS 3.14 [43].

As is clear from the analysis there are mainly small differences in the averages between centrelines and carriageways. The mean and median drive distance between all datasets is very similar despite the difference in detail between the two datasets. This highlights that the overall connectivity of the network is sufficiently preserved. Nevertheless, there are some individual routes that seem to have significant changes; we noticed that there can be decreases and increases in driving distance of one to six kilometres for individual routes, which is an important change for the specific routes. We believe that this highlights how specific cases can be greatly impacted by the difference in the details of a network and this should be taken into account based on the application for which the network is intended for. In addition, while the difference in the total distance of all routes might not be large as a percentage, the absolute values are still large enough to affect applications such as fuel consumption.

When analysing the results of the network analysis it is important to remember that the use case dictates the geometric level of detail required. In the case of grocery delivery routing, where precise distances are required to route the multiple delivery orders on a given day, carriageways may provide more realistic routing scenarios. Furthermore, in an industry where preparing to deliver on time requires models to account for the worst-case scenario vs. the best case scenario, carriageways provide a more conservative routing calculation. On the other hand, the processing time for routing on carriageways is larger than for centrelines, so if the goal of the use case is to do a more simple analysis (e.g. routing for scenic routes) then centrelines provide a faster solution.

Finally, the geometric level of detail can have implications for customers that live in-between two stores. Customer to store allocation could significantly affect creating retail catchment areas. As seen in Table 5, between 0.8% and 2.66% of customers experienced a change in the store they were allocated to. This can have implications given that different stores may have different product ranges which may impact customer shopping decisions e.g. the availability of international food products.

## Discussion

Our methodology reconstructs carriageways from initial centrelines based on the calculated width, and focuses mainly on solving the reconstruction of edges near intersections which proved to be the most complex. This is because there is diversity in the types of intersections, not only in topology (i.e. degrees of nodes and the number of carriageways of incident roads), but also in geometry. In our implementation we solved all cases that we consider to be most common and ensure that the algorithm can be applied to multiple datasets around the world, creating a robust enough output to be used for network analysis. Nevertheless, we did encounter some degenerate cases that we did not solve, especially in areas with irregular shapes of networks such as in The Hague, and we expect that this can occur when our methodology is applied to other areas. This is more prominent due to the nature of crowdsourced data, such as OSM, with which one is bound to encounter many surprising new road cases, deviant cases that break all the logic of the aforementioned methodology. This work was based on a balancing act of dealing with such deviant cases and adjusting for others, sometimes at a small cost to aesthetics (Fig 12).

Working with OSM proved to have its own influence on our research, due to its varying nature and the sometimes inconsistent manner in which roads are modelled. One of the issues we encountered was that sometimes dual carriageways were already drawn as individual lines in the original dataset (see Fig 8). Ideally, these existing carriageways should be identified and, when they connect, linked with our newly created carriageways. However, this is not a trivial process as it would require heuristics, in this case geometry, topology, and/or semantics (e.g. name of road), which could be used to group lines together. Not only is this not a trivial task, this would make the process more prone to errors. In addition, adjusting the geometry of these existing carriageways to join the new ones would be challenging in itself. Nevertheless, we expect that our statistics might be slightly skewed towards a lower number of dual carriageways, but given that these OSM modelled dual carriageways are not that common we do not think this is a big concern.

Another interesting observation about using OSM data is the dynamic nature of the data. Given that OSM is constantly being updated and our notebook always downloads the data directly from their service, we expect that running the exact same script multiple times will have different results. This is not only with respect to permanent changes in roads, but sometimes temporary changes as well. For example, in one instance of running our algorithm for

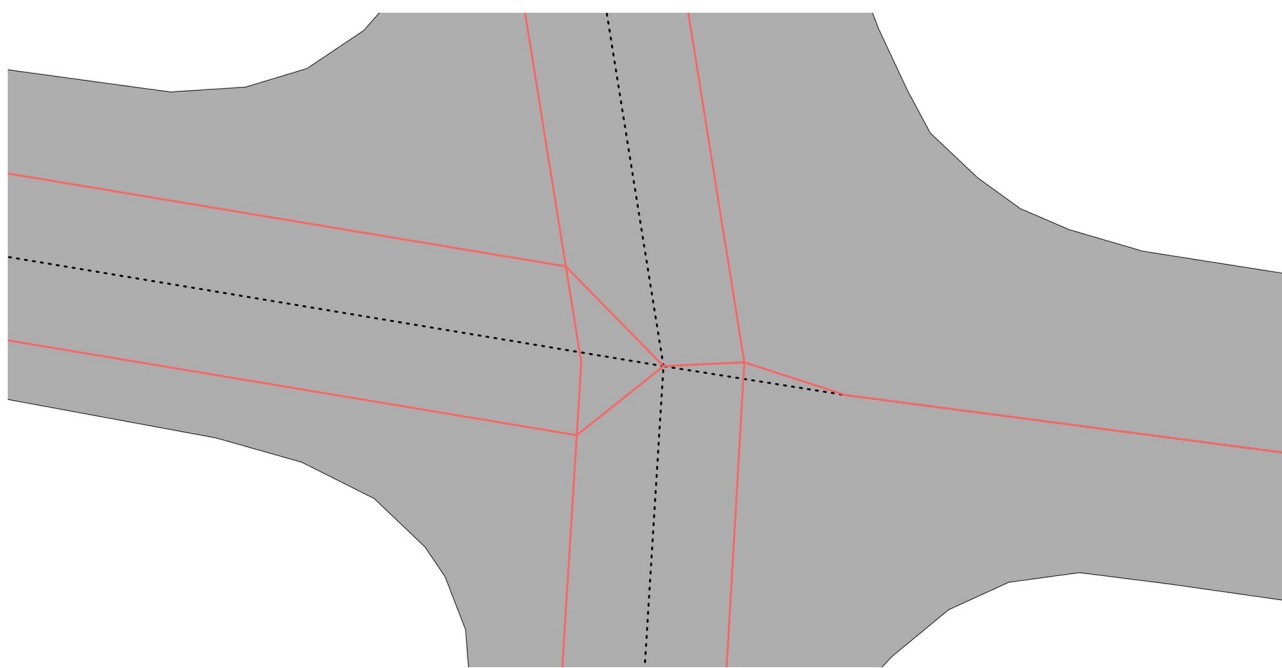

**Fig 12. Fixing overall topology in the network was prioritised over pure aesthetics.** Comparing Iteration 2 with Iteration 5 for a single to dual carriageway in an intersection.

Helsinki we realized that a whole neighbourhood was disjointed from the rest of the network because the one edge that was connecting them was a road temporarily closed at the time, which lead to this neighbourhood being an individual connected component and being excluded from the rest of the analysis.

An important part of our algorithm is the calculation of road widths, which affects the rest of the process. While it is essentially just a single step which was not solved by us, we encountered the complexity of the issue, and we realized it is not a straightforward or trivial problem to solve. Roads can be of variable width along their entire length and the way they are modelled in their areal representation can greatly influence this (Fig 13); how often roads are subdivided into polygons, intersections represented as one or multiple polygons, or how polygons are designed to represent physical areas can all greatly affect the outcome of a width calculation. For example, in The Hague we noticed that there were many road polygons with some rectangular holes in them (representing placeholders for transport objects such as traffic islands), so based on the road width calculation methodology these holes might have skewed the width of the entire road.

There is a high degree of variation in the resulting road widths for our datasets (Fig 14), which can be explained by the factors we mentioned before. The result is that sometimes the width computed was unreliable, especially close to intersections (Fig 15). In these cases, we tried to use topology to improve the outcome of the decision between dual and single carriageways, so if only the last part of a road section is defined as a dual carriageway while the rest is not then we mark it as a single carriageway.

With respect to the actual construction of edges in our algorithm, we noticed that the simple notion that edges are road segments can be misleading. This is because sometimes edges do not represent actual roads, but only serve the purpose of connectivity. For example, in cases of roads with two carriageways that are separated through a traffic island, there can be multiple

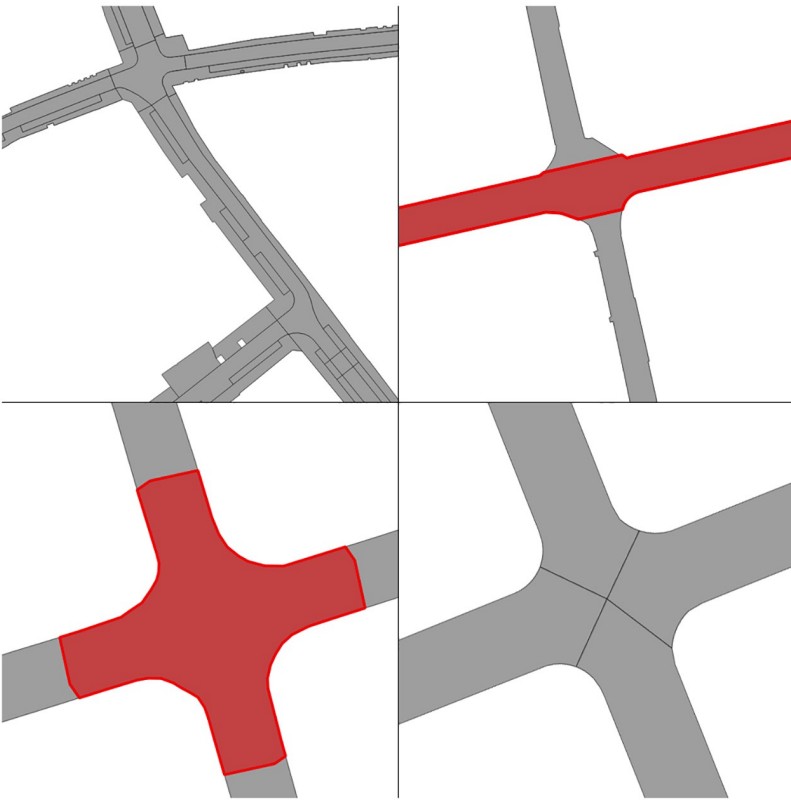

**Fig 13. The different approaches to modelling intersections, clockwise from top left: The Hague, Helsinki, Shawinigan, and Toronto.**

edges that represent a simple break of the traffic island. Our algorithm assumes these edges are roads and their width counts, but they only represent traffic flow instead of an actual road so their geometry (hence, their road width calculation) is irrelevant. We tried to exclude these edges from our calculation by applying a rule that when a short edge connects two nodes of degree higher than two, then we do not modify it.

Our methodology is iterative: for every processing step we are iterating through all edges and compute the outcome for the next step. This ensures that all edges are treated equally, meaning that they know the exact state of the rest of the network prior to this step. While this

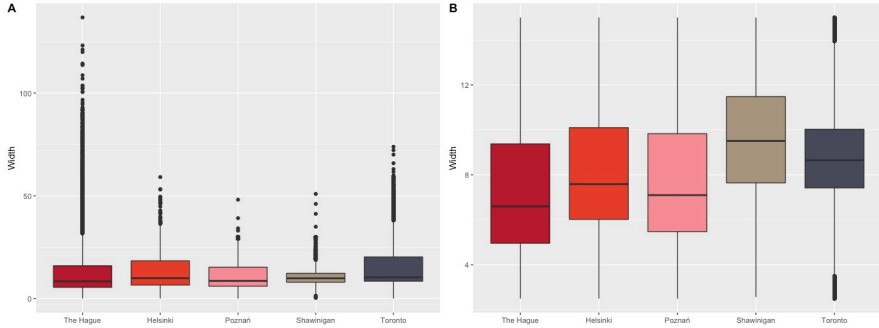

**Fig 14. The range of road width values per area in our study.** Figure A is all road width values. Figure B is road width values excluding outliers.

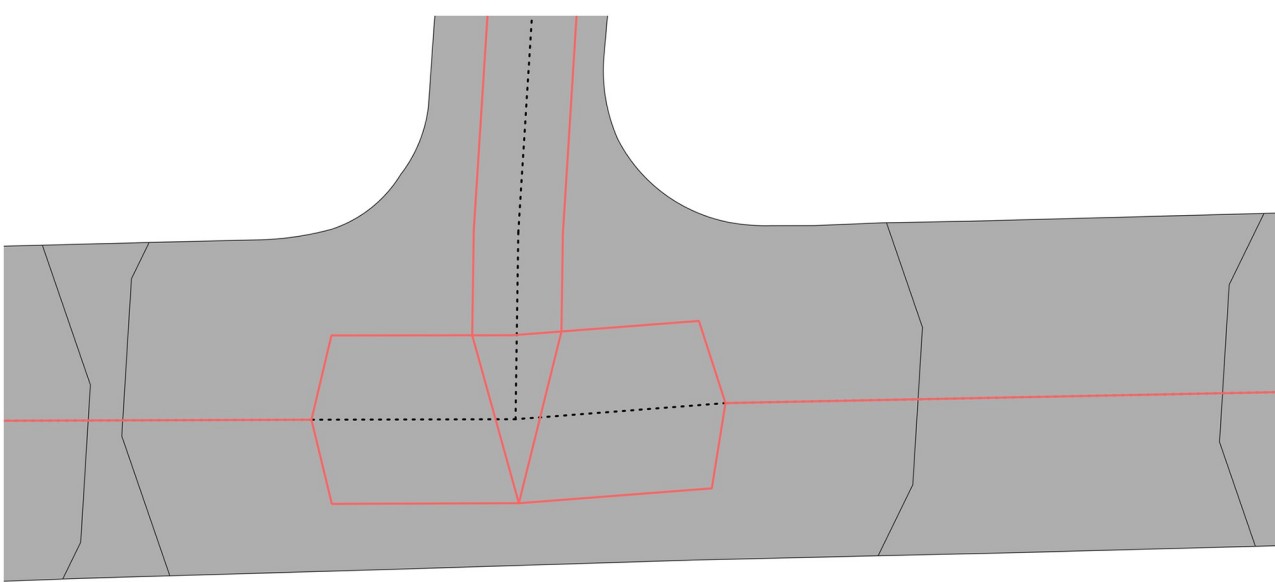

**Fig 15. A false "roundabout" generated based on the impact of the road width tolerance.**

guarantees consistency between the network (two edges in a similar environment get the same output despite the order that they are processed), it poses some limitations to our solution because sometimes when processing an edge we need to have information about how their neighbouring edges will be affected by this step as well. This creates a recursion that could become endless, therefore when this occurred we solved it by introducing multiple passes of the same step. For example, during the first iteration (carriageway creation) where we sometimes use the information of neighbouring edges' width to decide whether the current edge is a single or a dual carriageway, we encountered the problem that an edge might consider the neighbours to be single carriageway, but they ended up being dual (due to the influence of their neighbours). We decided to only do two passes for this calculation, because there is no finite solution to this problem.

Due to the iterative process, we were only able to use certain network characteristics with respect to every individual edge. Nevertheless, occasionally we had to rely on some geometric information (such as the angle between lines) to determine how to process the current edge. However, there is further information about the nodes themselves (i.e. the intersections) that could have further helped the algorithm. For example, knowing whether a three-way intersection is of a *T* or a *Y* shape could have replaced our existing angle calculations between edges and might have had better results. Although, this would be possible if the iterations were based on nodes, instead of edges.

Finally, flyovers are an important aspect to consider, especially when dealing with a 2D dataset. Flyovers impact network modelling because it is important to pay attention that nodes are not simply placed where two edges cross each other, where in cases of flyovers would indicate incorrect connectivity. We were concerned that flyovers would skew our road width calculations due to discrepancies we identified between grade separation classifications in linear and areal datasets. We were also concerned that identifying which polygons truly overlap with which road edges would be a difficult task. In the end, we found flyovers not to be an issue for our methodology because it was determined that OSM models flyovers as one-way lanes which we do not process.

## Conclusions

In this paper we presented a novel approach to automatically create an approximation of carriageways from centrelines of roads. We applied our algorithm to five areas in different countries (The Hague, The Netherlands; Helsinki, Finland; Poznań, Poland; Shawinigan, Canada; Toronto, Canada) using data from OpenStreetMap and local areal representations to compute the road widths (to determine which centrelines should be converted to dual carriageways). Our method demonstrates that it is possible to produce a useful network of carriageways based on a procedural process and heuristics.

In many studies, related to modelling carriageways, the generation method is often separate to the centreline generation. In other cases it is a purely manual method where various levels of detail are acquired by topographers or surveyors for a specific case study. Our study has examined how to generate carriageways from centrelines with the aim that the datasets can still be linked together for usage in various applications. Furthermore, many carriageway creation studies often focus on employing expensive technology, such as laser scanning, while we hope to harness the power of open data to generate the data easily. We believe that by automating this process and allowing it to run based on an open dataset that is available worldwide, such as OSM, we can democratise the process of creating carriageways consistently.

Our methodology was developed to be generic enough in order to ensure that it is geographically and application independent. Our main goal was to ensure that a valid and connected network is created, which can be easily adopted to work with any centerlines network dataset.

## Future work

Our current approach uses an areal representation to compute road widths in order to derive the number of carriageways for every road edge. We currently only focus on one road width value but it may be beneficial to look at several other values such as median, maximum, minimum, and standard deviation to see whether we can determine a more thorough road width calculation. Also, while we rely on having areal representations, these datasets are not widely available everywhere in the world, so our code could easily be adjusted in order to omit calculating road width and to instead derive the number of carriageways based on semantics (e.g. the type of road and its length).

Furthermore, our approach is currently tailored towards regions where forward-moving traffic is on the right-side of the road. While it would be fairly easy to adapt our methodology to work with left-side forward-moving traffic, due to the lack in availability of open areal representation datasets for such regions, we decided to leave this for future work. Adjusting the existing solution to work with left-side traffic will be trivial.

Finally, we briefly investigated the possibility of creating lanes, but it became evident to us that such a process comes with more challenges and different challenges than carriageway creation. This is because lanes are closer related to geometry, in the sense that they are often not symmetric (such as with carriageways) and they require much more detailed information especially with respect to road width calculations. Therefore, future work can focus on computing more detailed width information about roads and creating lanes based on the geometric characteristics of the areal representation of roads. Lane generation will also require an investigation into whether lanes should be generated from centrelines or from carriageways.

## Acknowledgments

We would like to thank Willem Hoffmans for providing a thorough explanation of his methodology and assisting with the adjustment of his work to make it work with our data.

## Author Contributions

**Conceptualization:** Stelios Vitalis, Anna Labetski, Hugo Ledoux, Jantien Stoter.

**Data curation:** Stelios Vitalis, Anna Labetski.

**Formal analysis:** Stelios Vitalis, Anna Labetski.

**Funding acquisition:** Jantien Stoter.

**Investigation:** Stelios Vitalis, Anna Labetski.

**Methodology:** Stelios Vitalis, Anna Labetski.

**Project administration:** Hugo Ledoux, Jantien Stoter.

**Resources:** Jantien Stoter.

**Software:** Stelios Vitalis.

**Supervision:** Hugo Ledoux, Jantien Stoter.

**Validation:** Stelios Vitalis, Anna Labetski.

**Visualization:** Stelios Vitalis.

**Writing – original draft:** Stelios Vitalis, Anna Labetski.

**Writing – review & editing:** Hugo Ledoux, Jantien Stoter.

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
