## [Decision Letter · Decision Letter 0]

12 Oct 2021

PONE-D-21-17971From road centrelines to carriageways - a reconstruction algorithmPLOS ONE

Dear Dr. Vitalis,

Thank you for submitting your manuscript to PLOS ONE. After careful consideration, we feel that it has merit but does not fully meet PLOS ONE’s publication criteria as it currently stands. Therefore, we invite you to submit a revised version of the manuscript that addresses the points raised during the review process. Please pay attention to the following critical points:1. You focus on the local match between the reconstructed and real carriageways.- What is the overall match between the reconstructed and real networks for the city neighborhood or region?- How you can compare the reconstructed and real patterns formally? - Are all matches/mismatches equally important?2. Is your study the first of this kind? If yes - claim that. If not - compare your algorithms to the existing ones3. Your algorithms are heuristic. How can you know that they are sufficient for resolving all OSM problems? Note that this remark is tightly related to the question about quality of your reconstruction at the regional level

We look forward to receiving your revised manuscript.

Kind regards,

Itzhak Benenson, Ph.D.

Academic Editor

PLOS ONE

2. Thank you for stating the following in the Acknowledgments/ Funding Section of your manuscript:

“The research leading to this paper has received funding from the European Research 680

Council under the European Union’s Horizon2020 ERC Agreement no. 677312 UMnD: 681

Urban modelling in higher dimensions”

“The research leading to this paper has received funding from the European Research Council under the European Union’s Horizon2020 ERC Agreement no. 677312 UMnD: Urban modelling in higher dimensions which was granted to JS.”

“No”

5. We note that Figures 2 and 11 in your submission contain [map/satellite] images which may be copyrighted. All PLOS content is published under the Creative Commons Attribution License (CC BY 4.0), which means that the manuscript, images, and Supporting Information files will be freely available online, and any third party is permitted to access, download, copy, distribute, and use these materials in any way, even commercially, with proper attribution. For these reasons, we cannot publish previously copyrighted maps or satellite images created using proprietary data, such as Google software (Google Maps, Street View, and Earth). For more information, see our copyright guidelines: http://journals.plos.org/plosone/s/licenses-and-copyright.

 a. You may seek permission from the original copyright holder of Figures 2 and 11 to publish the content specifically under the CC BY 4.0 license. 

Natural Earth (public domain): http://www.naturalearthdata.com/.

Additional Editor Comments (if provided):

Reviewers' comments:

Reviewer's Responses to Questions

**Comments to the Author**

1. Is the manuscript technically sound, and do the data support the conclusions?

Reviewer #1: Yes

Reviewer #2: Yes

2. Has the statistical analysis been performed appropriately and rigorously? 

Reviewer #1: Yes

Reviewer #2: I Don't Know

3. Have the authors made all data underlying the findings in their manuscript fully available?

Reviewer #1: Yes

Reviewer #2: Yes

4. Is the manuscript presented in an intelligible fashion and written in standard English?

Reviewer #1: Yes

Reviewer #2: Yes

5. Review Comments to the Author

Reviewer #1: 1. What is the novelty in the current work? Discuss how the present work adds value to the exiting works.

2. List out the main contributions of the current work.

3. Some of the recent work on intelligent transportation systems such as the following can be discussed in the paper: "CANintelliIDS: detecting in-vehicle intrusion attacks on a controller area network using CNN and attention-based GRU, Deep learning-based traffic safety solution for a mixture of autonomous and manual vehicles in a 5G-enabled intelligent transportation system".

4. Compare the results obtained with recent state of the art.

5. What is the computational complexity of the proposed approach?

Reviewer #2: 1. What are the limitations of the existing works that motivated the current work?

2. List out the main contributions of the current work.

3. Compare the results of the current work with the recent state-of-the-art.

4. Discuss the drawbacks of the current work in the conclusion.

5. Performance Comparison of proposed system with state-of-art literature is not provided any where in the article.

6. The disadvantages of the existing schemes must be discussed in tabular form.

7. The last paragraph of the Introduction must represent the structure of the paper in a more inspiring way.

6. PLOS authors have the option to publish the peer review history of their article (what does this mean?). If published, this will include your full peer review and any attached files.

Reviewer #1: No

Reviewer #2: **Yes: **Dr. Kadiyala Ramana

---

## [Author Response · Author response to Decision Letter 0]

24 Dec 2021

Thank you for your comments. Please, find our response to your suggestions below:

# Editor's comments

1. You focus on the local match between the reconstructed and real carriageways.

 - What is the overall match between the reconstructed and real networks for the city neighborhood or region?

 We are not ensuring a matching with the real-world carriageways, but instead we are confident enough that the carriageways we produce are a good approximation of the network. This is based on the fact that we trust the original datasets, both OSM as a reliable crowdsourced dataset of the roads network and the areal representations as they are of official institutional sources (i.e. local governments). Ensuring a high accuracy match between the reconstructed and real networks is out of the scope of this research. Instead, we are focusing on how to extract an approximation as representative enough as it can be based on the original data. The motivation of this research is to be able to automatically create a carriageways' network from easily accessible and actively maintained data. We amended our introduction to state that our work is about creating an approximation of carriageways to clarify this.

 - How you can compare the reconstructed and real patterns formally?

 We don't believe such a comparison can be achieved unless official sources of carriageway networks are provided. In this context and given the scope of our research, we evaluated our output's quality visually using satellite imagery and used our personal knowledge of some areas (Toronto and The Hague) to ensure that the output is, indeed, a good approximation.

 - Are all matches/mismatches equally important?

 The importance of mismatches between the reconstructed and real carriageways is heavily relying on the specific application for which the network is intended for. We set as the most important requirement for our work that the connectivity of the network is preserved. In order to ensure that our work is generic enough both application-wise and geographically we made most decisions prudently. As a result, we expect that some complex areas of the network, mostly flyovers or irregular and high-degree intersections, remain of lower detail in order to ensure that the connectivity persists. We highlight a few of these cases in our paper and we comment on the challenges that arise from them. We amended the manuscript to better reflect that, by making it explicit in the introduction and the conclusions that our methodology is about creating an approximation of carriageways. We, also, added a paragraph in the conclusions to clarify the features of our method that we prioritised the most.

2. Is your study the first of this kind? If yes - claim that. If not - compare your algorithms to the existing ones

 While we review other research about creating carriageways and lanes from satellite imagery, to our knowledge there is no other work that relies on open vector data to do so. Therefore, we believe that our current phrasing of a "novel reconstruction algorithm" in the introduction and conclusions fairly represents the uniqueness of this work. Nevertheless, we remain open to any other phrasing suggestions.

3. Your algorithms are heuristic. How can you know that they are sufficient for resolving all OSM problems? Note that this remark is tightly related to the question about quality of your reconstruction at the regional level

 Resolving any potential issues from OSM is out of the scope of this research. We developed our algorithm to be applicable in a wide variety of areas and, therefore, our focus was on ensuring that the output network is valid with respect to the input data provided. Our methodology can be easily adopted to work with other data sources. Nevertheless, we do trust OSM as a reliable source of data for our use case. We added a paragraph at the end of our conclusions to clarify the generic nature of our methodology and that it can be easily adopted to work with other centerline datasets.

# Reviewer 1 comments

1. What is the novelty in the current work? Discuss how the present work adds value to the exiting works.

While we review other research about creating carriageways and lanes from satellite imagery, to our knowledge there is no other work that relies on open vector data to do so.

2. List out the main contributions of the current work.

Our methodology works automatically. It, also, works with easily accessible and open vector data and is geographically and application neutral. We appended the conclusions to state these more explicitly.

3. Some of the recent work on intelligent transportation systems such as the following can be discussed in the paper: "CANintelliIDS: detecting in-vehicle intrusion attacks on a controller area network using CNN and attention-based GRU, Deep learning-based traffic safety solution for a mixture of autonomous and manual vehicles in a 5G-enabled intelligent transportation system".

This paper has nothing to do with our research, and we feel strongly that we do not need to include it in our paper.

4. Compare the results obtained with recent state of the art.

Based on our answer to question no 1, due to the vastly different nature of our research compared to other works and their use of proprietary data, we believe that it is impossible to provide a formal comparison between our work and any other studies related to carriageways reconstruction.

5. What is the computational complexity of the proposed approach?

We introduced a computational complexity analysis section in the paper to address this issue.

# Reviewer 2 comments

1. What are the limitations of the existing works that motivated the current work?

Current works use satellite imagery which is hard to find and expensive to acquire. In addition, there is no work about creating carriageways from vector data automatically.

2. List out the main contributions of the current work.

Our methodology works automatically. It, also, works with easily accessible and open vector data and is geographically and application neutral. We appended the conclusions to state these more explicitly.

3. Compare the results of the current work with the recent state-of-the-art.

Based on our answer to question no 1, due to the vastly different nature of our research compared to other works and their use of proprietary data, we believe that it is impossible to provide a formal comparison between our work and any other studies related to carriageways reconstruction.

4. Discuss the drawbacks of the current work in the conclusion.

We already discuss the drawbacks extensively in the discussion and future work, therefore we see no benefit of repeating them in the conclusions.

5. Performance Comparison of proposed system with state-of-art literature is not provided any where in the article.

We introduced a computational complexity analysis section in the paper to address this issue.

6. The disadvantages of the existing schemes must be discussed in tabular form.

We believe that our methodology is so vastly different to other works related to carriageways reconstruction, that such a table would not benefit the structure of the paper.

7. The last paragraph of the Introduction must represent the structure of the paper in a more inspiring way.

We find it hard to identify what an "inspiring way" means in this context. Unless some examples or clarification is provided regarding this point, we are satisfied with the existing phrasing of our structure of the paper.

---

## [Decision Letter · Decision Letter 1]

6 Jan 2022

From road centrelines to carriageways - a reconstruction algorithm

PONE-D-21-17971R1

Dear Dr. Vitalis,

We’re pleased to inform you that your manuscript has been judged scientifically suitable for publication and will be formally accepted for publication once it meets all outstanding technical requirements.

Kind regards,

Itzhak Benenson, Ph.D.

Academic Editor

PLOS ONE

Additional Editor Comments (optional):

Reviewers' comments:

Reviewer's Responses to Questions

**Comments to the Author**

1. If the authors have adequately addressed your comments raised in a previous round of review and you feel that this manuscript is now acceptable for publication, you may indicate that here to bypass the “Comments to the Author” section, enter your conflict of interest statement in the “Confidential to Editor” section, and submit your "Accept" recommendation.

Reviewer #2: All comments have been addressed

2. Is the manuscript technically sound, and do the data support the conclusions?

Reviewer #2: (No Response)

3. Has the statistical analysis been performed appropriately and rigorously? 

Reviewer #2: (No Response)

4. Have the authors made all data underlying the findings in their manuscript fully available?

Reviewer #2: (No Response)

5. Is the manuscript presented in an intelligible fashion and written in standard English?

Reviewer #2: (No Response)

6. Review Comments to the Author

Reviewer #2: (No Response)

7. PLOS authors have the option to publish the peer review history of their article (what does this mean?). If published, this will include your full peer review and any attached files.

Reviewer #2: **Yes: **Dr. Kadiyala Ramana

---

## [Editor Report · Acceptance letter]

10 Feb 2022

PONE-D-21-17971R1 

From road centrelines to carriageways - a reconstruction algorithm 

Dear Dr. Vitalis:

I'm pleased to inform you that your manuscript has been deemed suitable for publication in PLOS ONE. Congratulations! Your manuscript is now with our production department. 

Kind regards, 

on behalf of

Professor Itzhak Benenson 

Academic Editor

PLOS ONE